

# Monitoring landcover change and desertification processes in northern China and Mongolia using historical written sources and vegetation indices

Michael Kempf[1,2]

[1] Department of Archaeology and Museology, Masaryk University, Brno, Czech Republic, Arne Nováka 1, 60200 Brno, Czech Republic

[2] Physical Geography, Institute of Environmental Social Science and Geography, University of Freiburg, Germany, Schreiberstr. 20, 79085 Freiburg, Germany

*Correspondence to*: Michael Kempf (kempf@phil.muni.cz)

**Abstract.** Fighting land degradation of semi-arid and climate-sensitive grasslands are among the most urgent tasks of current eco-political agenda. Northern China and Mongolia are particularly prone to surface transformations caused by heavily increased livestock numbers during the 20th century. Extensive overgrazing and resource exploitation amplify regional climate change effects and trigger intensified surface transformation, which forces policy-driven interventions to prevent desertification. In the past, the region has been subject to major shifts in environmental and socio-cultural parameters, what makes it difficult to measure the extent of the regional anthropogenic impact and global climate change. This article analyses historical written sources, palaeoenvironmental data, and Normalized Difference Vegetation Index (NDVI) temporal series from the Moderate Resolution Imaging Spectroradiometer (MODIS) to compare landcover change during the Little Ice Age (LIA) and the reference period 2000-2018. Results show that decreasing precipitation and temperature records led to increased land degradation during the late 17th century. However, modern landcover data shows enhanced expansion of bare lands contrasting an increase in precipitation (Ptotal) and maximum temperature (Tmax). Vegetation response during the early growing season (March-May) and the late grazing season (September) does not relate to Ptotal and Tmax and generally low NDVI values indicate no major grassland recovery over the past 20 years.

## 1 Introduction

Climate and land cover change, heavy grazing, and agricultural as well as resource exploitation contribute significantly to land degradation and desertification processes in sensitive arid and semi-arid regions of the earth (Burrell et al., 2020; Cowie et al., 2011; Herrmann and Hutchinson, 2005; Pederson et al., 2001; UN, 1994; UNCCD, 2020; Vogt et al., 2011; Zhao et al., 2005). Particularly seasonal vegetation-cover plays a major role in the system's functionalities, affecting soil development, sediment deposition, water infiltration rate, and wind-driven erosion during high-cover and low-cover periods and within the spatial patterns of species differentiation (Aguiar and Sala, 1999; Lin et al., 2010). From a historical perspective, seasonal vegetation dynamics control transhumance (seasonal nomadic), socio-cultural, and socio-economic strategies, which nowadays have turned into politically motivated intensified sedentary patterns with regional and supra-regional environmental overstraining



that cause severe surface damage and sandy desertification through increased livestock grazing (Conte and Tilt, 2014; Geist and Lambin, 2004; Taylor, 2006; Zhao et al., 2005). The system-inherent interactions of land degradation, climate forcing,

anthropogenic impact, and ecological functionalities have for instance been emphasized by Geist and Lambin (2004), Herrmann and Hutchinson (2005), and Weber and Horst (2011), who suggest complexity not only within physically controlled but also socio-cultural driven systems (Geist and Lambin, 2004; Herrmann and Hutchinson, 2005; Weber and Horst, 2011). Particularly China's Inner Mongolia Autonomous Region and Mongolia have experienced massive political and economic development during the past decades, which enabled a strong transformation of the social, rural, and environmental life

(Cincotta et al., 1992; Glindemann et al., 2009; Jordan et al., 2016; Neupert, 1999; Wu et al., 2015), followed by governmental grassland restoration projects and policies to prevent desertification (Zhang et al., 2018; Zhang et al., 2020c). Although the modern anthropogenic impact on semi-arid environments of the region has recently been emphasized by a great many authors (Cao et al., 2013; Conte and Tilt, 2014; Harris, 2010; Jordan et al., 2016; Kakinuma et al., 2013), isolating the climatic signal affecting land degradation processes from human-induced landcover change still provides a great challenge (Harris, 2010). In

this article, historical written sources from 1688 AD were used to evaluate and reconstruct non-standardized climatic and land surface conditions during the Little Ice Age (LIA) in northern China and Mongolia and to compare them to modern landcover development, partly deriving from satellite hyperspectral imagery, Copernicus Global Land Service data, and historical and current climate proxy datasets. The historical data allows tracing surface vulnerability and land degradation processes through intensified wind-blown sand transport during the exceptional cold and dry period of the so-called Maunder Minimum of the

Little Ice Age (LIA) (Eddy, 1976; Eddy, 1983; Lean, 2000; Luterbacher et al., 2001; Shindell et al., 2001). The comparison of surface conditions during the late 17[th] century AD further enables the evaluation of current global climate change and the cross-validation of the strong anthropogenic overprint in semi-arid grasslands of northern China and southern Mongolia (Wu et al., 2015).

## 55  2 Environmental settings of the study area

The research area covers southern Mongolia, China's Inner Mongolia Autonomous Region, and parts of China north of Beijing (Fig. 1). Mongolia is a landlocked country north of China with a highly continental climate, extremely cold and dry winters, and short and hot summers (Struck et al., 2020). After long-term summer droughts, climate extreme events in winter (dzud) have frequently led to severe livestock perish, which caused strong socio-economic crisis due to a domestic product market

dependency of about 20 % (Angerer et al., 2008; Suzuki, 2013). Inner Mongolia is situated to the south at the northern margins of China and stretches from northeast to the northwest (37°30' – 53°23' N, 97° 10' – 125°50' E) (Chen et al., 2007). Most of the plateau-like elevated region is dominated by extensive grasslands, which are characterized by various types of steppe vegetation and a great sensitivity to climate and land-use change (Mu et al., 2013; Wang et al., 2013b; Xiao et al., 1995). The climate is monsoon-controlled, arid to semi-arid with cold and dry winters and hot and more humid conditions during summer

and towards the subhumid north-eastern part (Chen et al., 2007; Xiao et al., 1995). Different climatic zones have developed





under a prevailing continental climate with temperate and semi-humid conditions in the east and more semi-arid and arid conditions towards the western part (Wu et al., 2015). Mean annual precipitation ranges from 20-450 mm/a (long-term average < 200 mm/a (Zhang et al., 2020b)) with a peak during July and September (Fig. 2), which favours temporally limited plant growth (Wu et al., 2015). In combination with the high environmental vulnerability and climate extremes, the strongly

increased water consumption for agricultural purposes, animal husbandry, and particularly intensified governmental mining activities has amplified the drought risk of the region and led to a sequence of severely dry years (Huang et al., 2015; Pei et al., 2020; Suzuki, 2013; Wang et al., 2019b; Zhang et al., 2020b).

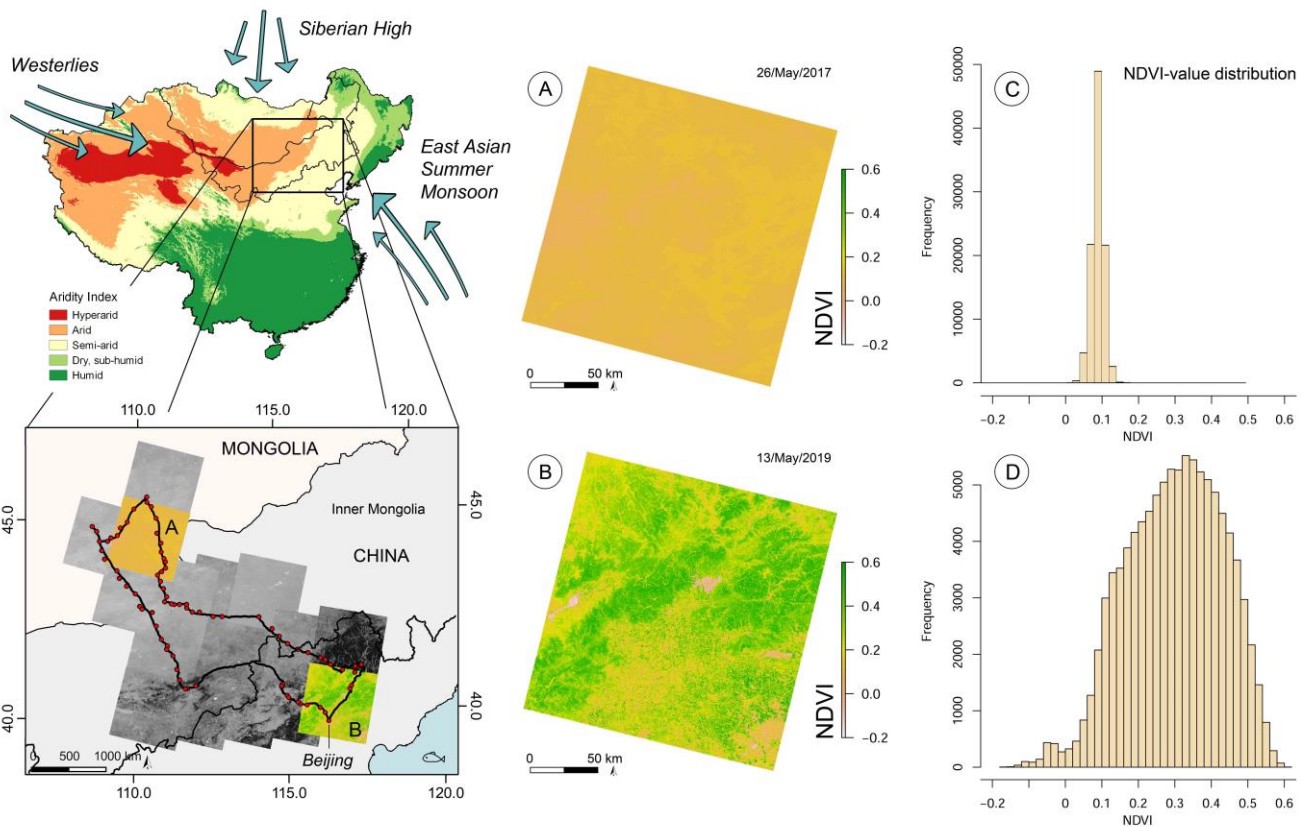

**Figure 1**: The study area in northern China and the southern part of Mongolia, covering approximately 433425 km². A set of Landsat-OLI-8 hyperspectral satellite images sets the extent of the reference area (USGS, last accessed 05 of January 2021). Vegetation indices (NDVI) allow for the evaluation of surface conditions through reflectance characteristics and eventually the differentiation into bare areas, vegetation-covered areas, and modern built-up (compare sections A and B). The threshold histograms (C, D) show the signal differences of two comparison sections in the north with extensive sand cover (A, C) and

the south over Beijing with mosaic vegetation cover and extensive built-up (B, D). The aridity index (AI) is based on the Global Aridity Index and Potential Evapotranspiration Climate Database v2 (Trabucco and Zomer, 2019).



## 3 Material and Methods

A comparison of historical written sources and environmental data attributes was carried out to evaluate modern desertification dynamics in northern China and southern Mongolia. Historical data derived from a day-by-day diary, which was documented

by the French missionary Jean-François Gerbillon, in the year 1688 during his employment at the imperial court at Beijing. While he was travelling from Beijing through Inner Mongolia and Mongolia during the period from the 30th of May to the 6th of October 1688, he documented climatic and surface conditions as well as general environmental, social, cultural, and political developments (Watts, 1739). From the documentation, a route model was reconstructed using terrain-dependent least-cost-path (LCP) analysis based on surface roughness and slope gradient (Herzog, 2014; Howey, 2011; Kempf, 2019) because

directional movement patterns were emphasized only in cardinal directions. A digital elevation model was downloaded from the United States Geological Survey (USGS, SRTM 1-arc-second/30 m resolution (Earth Resources Observation And Science (EROS) Center, 2017)) and resampled to 100 m grid size. The route model was calculated using QGIS and GRASS GIS and a cumulative friction surface and movement directions between the single stopping points. Around the reconstructed route, a 20 km buffer was created to visualize the historical environmental conditions within a suitable range. Climatic conditions were

classified from 1-6 with very hot (6), hot (5), warm (4), moderate (3), cold (2), very cold (1). Wind speed was estimated using cardinal directions and four classes from calm (0), breeze (1), wind (2) to storm (3). Surface conditions were distinguished according to the Copernicus Global Land Cover collection using the classification into bare/sand (0), herbaceous (1), shrub (2), cropland/grassland (3), forest/mixed (4), built-up (5), and water surface (6) (Buchhorn et al., 2020a; Buchhorn et al., 2020b). Climatic and surface data was interpolated within the 20 km range along the route and compared to modern land cover

data and temperature variations using the Copernicus Global Land Cover 100 m collection 3 from 2019 (Buchhorn et al., 2020a), a temporal series of Landsat-OLI-8 satellite imagery and vegetation indices (NDVI = NIR-Red / NIR+Red) (Tucker, 1979) covering cloud-free images from May to October 2013-2020 (Earthexplorer, USGS), precipitation and temperature comparison datasets for the period 1970-2000 (Fick and Hijmans, 2017) and 1961-2018 (University Of East Anglia Climatic Research Unit et al., 2019) (Fig.2). The aridity index (AI) is based on the Global Aridity Index and Potential Evapotranspiration

Climate Database v2 (Trabucco and Zomer, 2019) and was cropped to the study area extent. Because desertification and land degradation processes are strongly connected to windspeed and surface erosion, the historical wind direction and intensity was compared to modern data extracted from the Global Windatlas (Technical University of Denmark, 2020).

Because of the temporal variation of the surface and the climatic conditions, the study area was differentiated into equal monthly sectors using Voronoi polygons. Monthly maximum temperature and precipitation totals were plotted in each sector

to trace the seasonal variability in Inner Mongolia and southern Mongolia over the summer period. Together with a temporal series of Landsat-OLI-8 images, which were reclassified using R software, the vegetation variability can be evaluated during the summer maximum precipitation period. To cross-check the historical data analysis, a set of comparison climate data was acquired from the National Centers for Environmental Information (NOAA, last accessed 05 of January 2021) and plotted using a locally estimated scatterplot smoother (LOESS) and R software (Cleveland, 1979) with smoothing parameters 1, 0.5,



and 0.3. Precipitation reconstruction is based on tree-ring chronologies for two regions in China (Yi et al., 2010). Temperature anomaly and reconstructed temperature is based on a stalagmite from Shihua Cave, Beijing, China and instrumental meteorological records (Tan et al., 2003). Long-term streamflow variation of the River Kherlen derived from the dataset by Pederson et al. (2001) and spatial and temporal tree-ring replication and nested model methods by Davi et al. (2013) (Davi et al., 2013; Pederson et al., 2001). Streamflow variation of the river Selenge in Mongolia is based on tree-ring-width

chronologies (Davi et al., 2006) and precipitation reconstruction in north-eastern Mongolia derived from tree-ring-width data (Pederson et al., 2001).

To trace vegetation behaviour over the growing season, a temporal series was created from Moderate Resolution Imaging Spectroradiometer (MODIS) Terra MOD09A1 version 6 imagery and the spectral reflectance bands 1 (red) and 2 (near infrared), which provide a spatial resolution of 500 m with an 8-day return period (tiles h26v04; NASA;

https://cmr.earthdata.nasa.gov/, last accessed 12[th] of January 2021) (Vermote, 2015). The downloaded data covers the period March-May (MAM) and September 2000-2018 in Inner Mongolia, which was compared to climatic variables (Feng et al., 2016). MAM represents the early phase of the growing season in Inner Mongolia (Gong et al., 2015) and September is considered the end of the extensive livestock grazing period (Ren et al., 2012). Consequently, both periods are suitable to monitoring long-term vegetation growth behaviour. The images have been reprojected to the local WGS84 UTM zone 49N

and the NDVI was calculated from the bands 1 and 2 using the equation described above. Cloud-cover affects the spectral reflectance signal of the optical sensors and has been cross-checked using RGB-image calculations from the channels (1, 4, 3). Consequently, all negative NDVI values and values < 0.05 were removed from the dataset. The images were clipped to a rectangle covering a representative section of the research area and a set of 45 regular points were created with a distance of 25 km. For the period MAM, four images/month were assigned to each point and plotted with a locally estimated scatterplot

smoother to visualize trends in physical plant conditions during the growing season in Inner Mongolia. September images range from the 6[th] to the 13[th] and the 5[th] to the 12[th] of the month (Kawamura et al., 2005; Wang et al., 2019a; Wei et al., 2020; Zhou et al., 2017). The section covering the 45 regular points was cross validated with 127 regular points covering the entire extent of the MODIS scene over September 2000-2018. The NDVI-values of the study area section in Inner Mongolia behave like the values of the large-scale section of the MODIS scene, which rendered the section representative for further analysis.

It was not possible to enlarge the study area to the entire MODIS scene due to extensive cloud cover during the early growing season in parts of northern China. Monthly maximum temperature (Tmax) and monthly total precipitation (Ptotal) were extracted for the 45 regular measurements points over the period 2000-2018 in MAM and September. Correlation between monthly NDVI-value, Tmax, and Ptotal was tested using Pearson's correlation, which measures a linear dependence between two variables and plotted using the ggpubr R-package.







**Figure 2**: Monthly total precipitation in Inner Mongolia (red) and Mongolia (blue) over the period 1961-2018 deriving from 20 random sites based on the CRU TS4.03 (Climatic Research Unit Time Series) high resolution gridded data of month-by-month variation from I. C. Harris and P. D. Jones (University Of East Anglia Climatic Research Unit et al., 2019), last accessed 05 of January 2021 via http://data.ceda.ac.uk/badc/cru/data/cru_ts/cru_ts_4.03/data/pre. Data filtered with a locally estimated
scatterplot smoother (LOESS) and R software (Cleveland, 1979) with a smoothing parameter of 0.1.

## 4 Results and discussion

Today, the transition zone from northern China to Inner Mongolia and Mongolia is characterized by a pronounced landcover gradient from moderate forested areas in the south and the south-east to increasingly semi-arid and arid conditions towards the
Mongolian Plateau and the extensive grasslands of Inner Mongolia and Mongolia (Fig. 3). The Copernicus 2019 data shows the landcover sequences and compares topographical features for each landcover class in the study area. Forested zones are mostly abundant in lower elevated areas of the subhumid belt north of Beijing. With increasing elevation, semi-arid conditions prevail, which favour patterns of herbaceous grassland and shrubs. Towards the north-west, extensive sandy lands occur with a mean elevation of about 1000 m a.s.l. Croplands are frequently interspersed into semi-arid grassland patches, which points
towards an active anthropogenic land-use. Figure 3 further visualizes the historical route patterns and the daily camps of the travels of Pater Gerbillon from 1688. From these route reconstructions, a cross-validation of hermeneutic sources and modern landcover and climate data was derived.



**Figure 3**: Landcover in the study area in 2019 compared to topographical elements and the historical route reconstruction from
1688 (Buchhorn et al., 2020a; Earth Resources Observation And Science (EROS) Center, 2017; Watts, 1739).

### 4.1 Historical landcover change and environmental reconstruction

A route has been reconstructed from the historical data, which starts at Beijing on the 30<sup>th</sup> of May 1688. Travelling to the north-west, the group crossed the mountain range north of China's capital before they entered the Mongolian Plateau in the
first week of June. According to topographical changes, the landcover develops rapidly from forested to sparsely forested and herbaceous surfaces, and finally to shrubland, grassland, and steppe vegetation from the south-east to the north-west (Fig. 4). During the first week, the author reported from small-scale agricultural crop production, which were interspersed into the extensive grasslands. Furthermore, he frequently highlighted the absence of field systems despite the high suitability for intensive agricultural utilization of the region, which was connected to the prior nomadic lifestyle of the local population



(Chen, 2015; Cui et al., 2019; Neupert, 1999; Wu et al., 2015). The weather conditions were generally very dry with very high temperatures during the beginning of June and more humid and moderate conditions in the second half of the month when the group turned to the south-west and crossed extensive sandy grasslands with poor vegetation cover and bare hilltops (Fig. 5). On the 17th of June, they reached Hohhot, which is located on the Tumuochuan Plain and surrounded by the Hetao Plateau to the south and the Daqing Mountain to the north (Fan et al., 2016). The average temperature (6.7 ° C) and the annual total

precipitation (400 m/a) at Hohhot are low, which supports a semi-arid steppe climate (Fan et al., 2016). Both, the modern data and the historical landcover reconstruction indicate forested zones in the area, which are probably connected to the lower elevation compared to the surrounding elevated hills. After the 17th of June, the group moved northwards, and the vegetation cover declined according to an increase in aridity and windspeed (Fig. 6). After the 27th, Gerbillon reported from bare lands with no vegetation but loose sand coverage until the 31st of July 1688. In contrast to the 2019 landcover and the NDVI temporal

series, which was calculated from Landsat-OLI-8 satellite imagery, the 1688 landcover reconstruction shows increasing herbaceous vegetation patterns after the 31st of July despite continuously very dry conditions during August. The surface description of the following period until the 17th of September differs again considerably from modern landcover data. There is a signal towards more herbaceous and shrubby vegetation and increasing agricultural exploitation during the late 17th century. During the rest of September, the group travelled continuously to the east and entered the forested mountains around

the 22nd. Compared to the modern data, the extent of the forested areas reached further to the north and the north-west, which is most likely linked to strong modern human impact, forest management and climate change during the first half of the 20th century (Yu et al., 2011). Climatic conditions were reported to be very dry and extremely cold during October 1688, which aligns with the climatic tendency towards a drier and colder period around 1700 AD and the climate depression during the Maunder Minimum of the LIA. From the palaeoenvironmental reconstructions, 1688 can be considered an extremely

anomalous year compared to the long-term average and marks the transition into a generally colder and drier phase that lasts until about 1715 AD.

These findings are also supported by the modelled windspeed and the average wind direction from 1688 compared to modern data (Technical University of Denmark, 2020). The diary reports from continuously blowing wind with local extreme events and massively increased sand transport and dune activity. The reconstruction indicates an increase in windspeed and a general

change in wind direction during the LIA (Fig. 6). Today's landcover is even more vulnerable to wind erosion due to the locally decreased vegetation cover and intensified overgrazing, which reactivate sand depositions and enable dust transportation and dune development. The accumulation of coarse particles further supports the degradation of cropland and pasture (Hoffmann et al., 2011; Jiang et al., 2016; Li et al., 2005; Zhao et al., 2006). Sand and dust storms over Inner Mongolia are not only enhancing erosion and accumulation of fine-grained particles in the semi-arid steppe region but also lead to the transportation

of dust and high concentrations of particulate pollutants into the area of Beijing (Guo et al., 2004; Hu et al., 2016; Yang et al., 2015).





**Figure 4**: Landcover in 1688 as reported from the travels of Pater Gerbillon (Watts, 1739). During May to October 1688, the missionary crossed large parts of northern China and southern Mongolia and turned back to Beijing after half a year. He reported from weather as well as surface and vegetation conditions and enabled the reconstruction of past landcover conditions during the Maunder Minimum of the Little Ice Age.



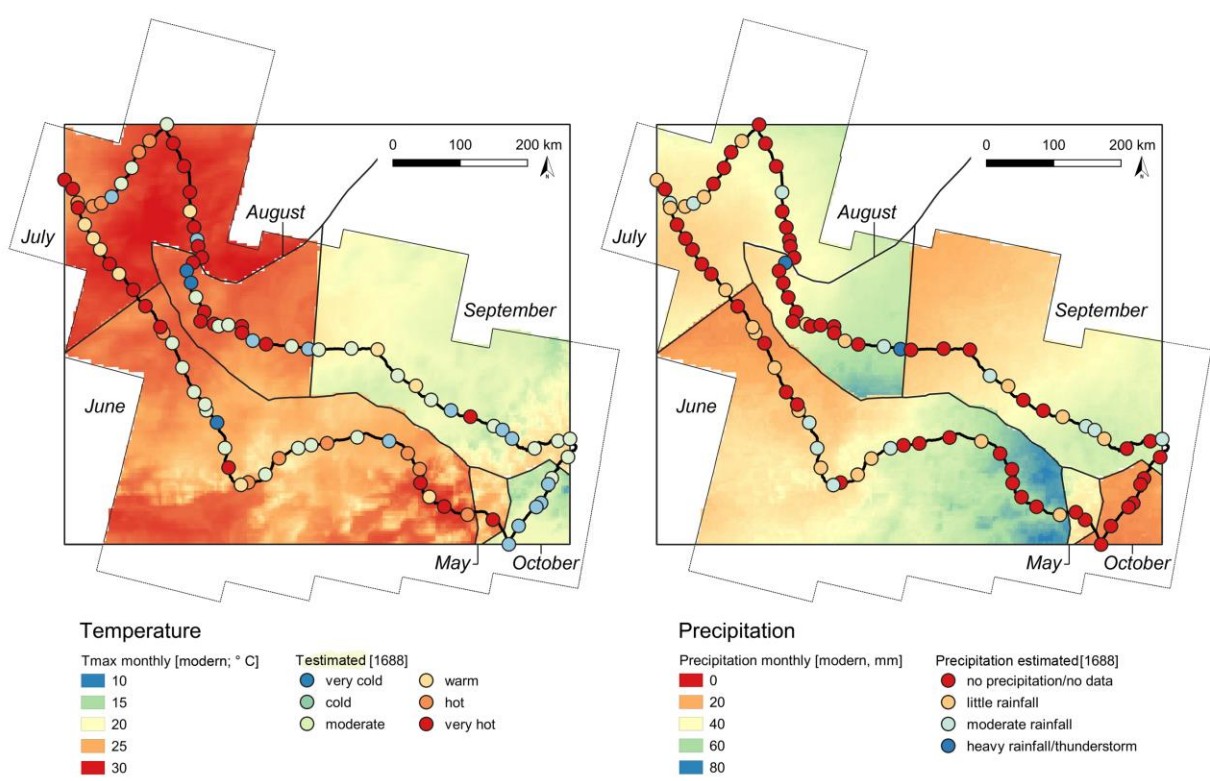

**Figure 5**: Average monthly temperature and precipitation grid in the study area from May to October. The reconstructed climatic conditions from 1688 were plotted on top of the modern dataset to highlight differences and similarities during the Little Ice Age (University Of East Anglia Climatic Research Unit et al., 2019; Fick and Hijmans, 2017; Watts, 1739).





**Figure 6**: (A) Modern average monthly windspeed diagrams of the study area from May to October (Technical University of Denmark, 2020). (B) Reconstructed wind directions for June – September 1688. May and October did not provide sufficcient



data. The reconstructed windspeeds and directions were plotted with modern datasets to underline the current trend towards increased windspeed, north-western wind direction, and sand transport and enhanced desertification risk.

To cross validate the hermeneutic proxy, historical and current environmental conditions in the study area were evaluated using palaeoenvironmental proxies from different data sources (Davi et al., 2006; Davi et al., 2013; Pederson et al., 2001; Tan et al., 2003; Yi et al., 2010) and interpolated datasets from modern weather stations (University Of East Anglia Climatic Research Unit et al., 2019; Fick and Hijmans, 2017). As described above, monthly total precipitation in Inner Mongolia and Mongolia does not show a significant decline over the period 1961-2018 (Fig. 2). Palaeoclimate reconstructions further reveal

no trend in precipitation decrease but rather a tendency towards greater annual variation (Fig. 7). As expected from global climate change models (Alverson et al., 2003), the reconstructed temperature increased significantly during the past 500 years. Reconstructed streamflow runoff in Mongolia is connected to the precipitation variability, and an increase of rainfall triggers peaks in runoff totals. However, there is no observable negative trend in river runoff during the past 400 years. It is particularly striking that the year 1688, which marks the peak of the Maunder Minimum of the LIA, can be characterized as an exceptionally

cold and dry year compared to the long-term average. The period 1675 to 1715, which is characterized by a sunspot minimum and decreased solar activity, is clearly visible in the temperature reconstruction from tree-ring width and peaks around 1700 (Tan et al., 2003). The minimum could have affected the East Asian Summer Monsoon (EASM) according to the 11-year solar cycle (Chen et al., 2020; Zhang et al., 2020a). The precipitation record of Urgun Nars (Pederson et al., 2001) shows a negative trend during the Maunder Minimum, which points towards a decreased humidity transport into semi-arid and arid Mongolia

caused by a potential decline of the EASM (Lan et al., 2020).



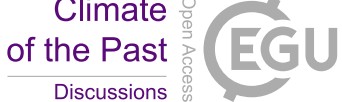

**Figure 7**: Paleoclimate reconstruction from six study sites in China and Mongolia. Raw data acquired from NOAA - National Centers for Environmental Information (last accessed 05 of January 2021) and plotted by the author using a locally estimated





scatterplot smoother (LOESS) and R software (Cleveland, 1979); smoothing parameters 1 (blue line, plot C, D), 0.5 (green
line), 0.3 (red line). The year 1688 is marked with a vertical, the average value with a horizontal blue line. (A, B): Precipitation
reconstruction based on tree-ring chronologies for two regions in China (Yi et al., 2010). (C): Temperature anomaly and (D):
reconstructed temperature based on a stalagmite from Shihua Cave, Beijing, China and instrumental meteorological records
(Tan et al., 2003). (E): River Kherlen long-term streamflow variation based on (Pederson et al., 2001) and spatial and temporal

tree-ring replication and nested model methods (Davi et al., 2013). (F): Streamflow variation of the river Selenge in Mongolia
based on tree-ring-width chronologies (Davi et al., 2006). (G): Precipitation reconstruction in north-eastern Mongolia based
on tree-ring-width data (Pederson et al., 2001).

## 4.2 Environmental transformation and land degradation

Landsat-OLI-8 and MODIS hyperspectral imagery were used to monitor vegetation canopy changes and surface
transformations linked to climate change and anthropogenic overstraining. Vegetation indices of subsequent months and years
and on various spatial scales allow for temporal in-depth observations of physical plant behaviour or drought periods and are
a common tool in remote sensing of ecological and climatic processes (Fensholt and Proud, 2012; Gu et al., 2009; Kempf and
Glaser, 2020; Ren et al., 2018; Wei et al., 2020). A set of cloud-free Landsat-8 images spanning the period May-October 2013-

2020 shows the monthly differentiation of vegetation cover in the study area (Fig. 8). As derived from the Copernicus landcover
data and the aridity index, the north-western part of the study area reveals significantly low plant physiological activity and
bare and sandy areas. This accounts for most of the transitional zone between China and Mongolia although the temporal series
covers the major precipitation period from July to September. From the NDVI-value classification, a section of the study area
was chosen that covers a transition zone characterized by shrubland and herbaceous vegetation, which is still utilized for crop

production. The long-term MODIS NDVI temporal series from 2000-2018 shows no major trend over the three-month early
growing season from March to May (Fig. 9). The early growing season is the most important period to determine vegetation
dynamics (Gong et al., 2015; Ren et al., 2012). Although Ren et al. (2012) highlighted the variability of rainfall and temperature
as the most important driving factor of vegetation dynamics in Inner Mongolia, the results from the NDVI long-term series
cannot confirm a direct relationship between climatic variability and NDVI between March and May. Maximum temperature

in the study area increased strongly during the past 20 years (Tong et al., 2017) and no particular trend but multiannual variation
can be observed in the precipitation totals (Zhang et al., 2019). No significant relationship between NDVI and maximum
temperature (Lauenroth and Sala, 1992) and NDVI and precipitation development can be detected from Pearson's correlation
test (Lu et al., 2019). In this context, Tong et al. (2017) report similar results from the eastern part of Inner Mongolia, where
NDVI-values increased from 1984 to 2013. The correlation between NDVI and precipitation and temperature, however, cannot

be confirmed for the study area, which is most-likely connected to seasonal variation in precipitation totals in eastern Inner
Mongolia and the annual cycle (Liu et al., 2019; Tong et al., 2017; Zhang et al., 2019).





These results confirm the complexity of aboveground net primary productivity (ANPP) in grasslands and annual, interannual, seasonal, and previous-year precipitation variability as reported from Inner Mongolia and North America shortgrass steppe
(Knapp and Smith, 2001; Lauenroth and Sala, 1992; Li et al., 2018; Ma et al., 2010; Oesterheld et al., 2001; Ren et al., 2018). Furthermore, there is a stronger spatial gradient of the sensitivity to and the relationship between precipitation and maximum temperature in desert steppe vegetation than in the subhumid forest zones (Li et al., 2018; Na et al., 2018). That strongly points towards the anthropogenically induced origin of local desertification processes through grazing activity after the growing season, which amplifies the vulnerability to global climate change of Inner Mongolia's grassland and steppe vegetation.
Grazing activity in Inner Mongolia's grasslands temporally peaks from July to September, when plant growth terminates (Ren et al., 2012). The comparison of the MAM NDVI-values with the September NDVI-values reveals only a slight increase in vegetation cover but a general low physical plant condition. Although the vegetation indices in this study remain very low during the early growing season and do not show a strong increase during grazing season, there is a tendency towards vegetation recovery, higher precipitation totals, and increasing maximum temperatures – particularly during the MAM period.
Increasing maximum temperatures during MAM would therefore advance the spring phenological phases, however, decreased soil moisture would delay them (Huang et al., 2019). During the late grazing season maximum temperature shows a slight decline over the period 2000 to 2018, which correlates better with the physical plant conditions while the total precipitation remains low and does not correlate with a potential vegetation recovery.

Recent research results have shown that land degradation is reduced considerably during the past 20 years and that desert
extent reduction is not primarily caused by a reduction in human grazing activity but rather by an increase in precipitation (Guo et al., 2020). Guo et al. (2020) report from a decrease in active desertification, however, this is mostly restricted to the more sub-humid northern part of Mongolia and the eastern parts of Inner Mongolia and does not totally affect the transitional zone between Inner Mongolia's and Mongolia's grasslands (Guo et al., 2020). Land degradation in northern China has increased constantly since the 1950's and peaked during the 1970's and 1980's after when it decreased continuously (Feng et
al., 2016). Since then, it has been frequently discussed whether desertification processes are caused by anthropogenic overstraining and particularly overgrazing activity or by climate change phenomena (Guo et al., 2020; Miao et al., 2015; Na et al., 2018; Wang et al., 2013a). Climate change and natural response cycles have been determined to trigger land degradation and desertification at variable scales. The complexity of human-natural global change and the feedbacks are particularly visible in semi-arid, climate-sensitive areas of the earth where strong local anthropogenic impacts on short temporal scales lead to
massively increased mobility of both humans and livestock after surface transformation and consequently the degradation of neighbouring boundary zones (Feng et al., 2016). That leads to a rapid decrease of the ecosystem's functionality and landscape connectivity and enhances surface degradation through aeolian processes and dust accumulation. Subsequent dry years, hot drought phenomena, and rising temperatures during the early growing season and summer can further strengthen the vulnerability of the steppe grassland in northern China (see Fig. 9) (An et al., 2020; Li et al., 2020).






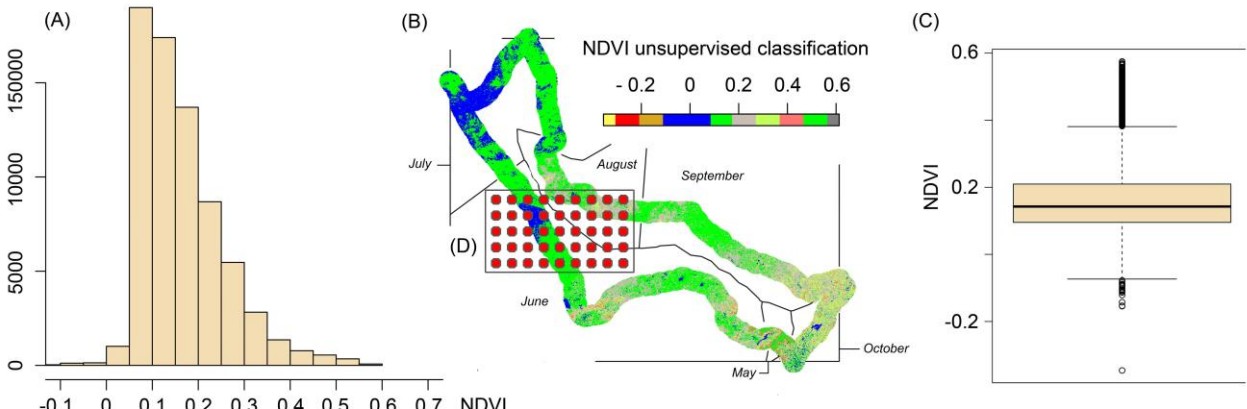

**Figure 8**: NDVI temporal series derived from cloud-free Landsat-OLI-8 images covering May to October 2013-2020 plotted over the study area and clipped to the 2 km buffer around the reconstructed route from 1688. Histogram of the NDVI-value distribution within the buffer region (A), unsupervised classification of the NDVI values (B), and boxplot of the value frequency between 0.1 and 0.2 (C). The rectangle shows the extent of the 45 regular points that were computed to produce the MODIS and the temperature and precipitation time series (D) (see Fig. 9 and 10).







**Figure 9**: MODIS 19-years long-term NDVI-value distribution, Tmax, and Ptotal series for 45 regular points over the growing
season March-April-May (MAM) in Inner Mongolia (see Fig. 8). The time series are plotted with a locally estimated scatterplot
smoother (LOESS) and the smoothing parameter 0.3. Pearson's correlation for March, April, May: Relationship between
NDVI and Tmax; NDVI and Ptotal. There is no correlation between variations in precipitation, temperature, and vegetation
growth behaviour over the growing season during the period 2000-2018.







**Figure 10**: NDVI, Tmax, and Ptotal during the second week of September over the temporal series 2000-2018 in the study area. Relationship between NDVI and Tmax, and NDVI and Ptotal using Pearson's correlation. There is no correlation between NDVI and precipitation but rather between NDVI and declining temperature.

**5 Conclusion**

Northern China's and Mongolia's climate sensitive, semi-arid regions experienced severe desertification during the 20[th] century, mostly linked to massively intensified livestock grazing activity, resource exploitation, and agricultural crop production, which increased water consumption and enhanced surface erosion. During the past decades, however, China's policy-driven decision-making processes pushed local to regional programs to prevent land degradation and stabilize sandy
areas and grasslands in order to decrease the potential of future soil erosion, surface transformation, and dust transport – a crucial factor, particularly in the context of Beijing's high vulnerability to increased numbers of sandstorms. The actual cause and effect of desertification processes, however, is still heavily debated and a great many research discusses whether local surface transformations are triggered by regional climate change feedbacks or whether they are connected to anthropogenically induced system transformation. It is a matter of fact that both are rooted in the human impact on the landscape functionalities,
and land degradation and desertification mirror only the ultimate collapse and loss of resilience to withstand enhanced climatic or human pressure.

From the results of this article, no direct relationship between environmental change and land degradation processes can be derived. That questions the actual impact of climate change on the semi-arid regions of northern China and southern Mongolia. No significant land degradation tendency is seen from the long-term NDVI time series, but rather slight increase in precipitation
and a significant increase in maximum temperature. On the other hand, this also indicates no tendency towards restored vegetation patches or limited livestock grazing. Even though China's government seeks to restore extensive grasslands to maintain the regional population and one of the earth's largest and growing livestock, seasonal variability in precipitation and increasing maximum temperatures and drought risk during the growing season will enhance the climatic pressure on semi-arid landscapes.
In this article, historical climatological analysis and hermeneutics were merged with long-term palaeoenvironmental data and remote sensing techniques to reconstruct surface transformation and climate development during the LIA and compare land degradation processes to modern desertification trends in semi-arid northern China and Mongolia. For this reason, written sources from 1688 were evaluated to extract a temporal series of landcover conditions for the period May to October 1688 – a year that falls into a period of reduced sunspot activity and solar energy flux during the LIA. Palaeoenvironmental proxy have
shown that precipitation and temperature records decreased during the Maunder Minimum (1675 to 1715 AD). 1688 is reported to peak with extreme climatic conditions by tree-ring-width and stalagmite composition analysis. According to written sources, the year was characterized by extremely low temperatures during late grazing season of September and the onset of October and extremely dry conditions and severely high temperatures during summer rainy season, which caused massive perish of



livestock in the region. From the reconstructed landcover classification and the comparison to modern Copernicus landcover
data from 2019, it becomes visible that degraded bare sandy areas advanced under current climate and anthropogenic change
in the region. Even though the evaluation of hermeneutically deduced historical environmental data remains strongly
subjective, it represents a useful source to measure the dimension of human landcover change on the long-term scale. This is
particularly important in grasslands and steppe vegetation areas, which are the earth's most climate-sensitive resources.

The author declares no conflict of interest.

This research received funding from Operational Programme Research, Development and Education - Project
„Postdoc2MUNI" (No. CZ.02.2.69/0.0/0.0/18_053/0016952), Masaryk University, Brno, Czech Republic.

Acknowledgements

I am very grateful to Alberto Reyes for comments on the initial submission of the manuscript. Furthermore, I would like to
thank…..

This research was supported from Operational Programme Research, Development and Education - Project „Postdoc2MUNI"
(No. CZ.02.2.69/0.0/0.0/18_053/0016952).

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
