# Peer review of "Monitoring landcover change and desertification processes in northern China and Mongolia using historical written sources and vegetation indices"

_Climate of the Past, 2021_

## Author Comment (AC1)

*Thank you very much for your comments and critics on the manuscript. In the following, I tried to answer the issues you raised during the review process. Please find my answers highlighted in italics. I am looking forward to further discussing your interesting suggestions! It seems like you uploaded your review twice – as anonymous reviewer and again under your clear name. I will upload only one response, if this is fine with the editor.*

Kempf uses current climate data, satellite NDVI, paleoenvironmental proxies and historical written sources to compare and evaluate the landcover change and land degradation in the Mongolian Plateau. The major source of data for evaluating the land degradation is the NDVI values for current time period, although it is unclear what indices where used to evaluate the land degradation for the historical time period.

*Reconstructing past surface cover over large areas by means of historical data (mostly written sources and spot-light palaeoenvironmental proxy) cannot generate indices in the very sense of the term 'index' – and particularly taking into account that NDVI by design shows values between -1 and 1 on a continuous scale. The evaluation of historical data cannot produce continuous data but rather discrete values, such as bare (0), grassland (1), forest (2). That is also a major limitation in terms of quantitative statistics because discrete values were not supported by some of the packages implemented in R software.*

I commend the author for putting so much effort in deriving meaningful information from historical written records.

*Thanks!*

Main findings of the study were that vegetation change was not related to any climate variables and therefore the authors attributed grassland degradation to increasing livestock density in the region.

*That is probably a very condensed summary of the output and I would rather distinguish the two chronological timeframes, which support the results. The historical landcover change was certainly triggered by LIA temperature decline and local to regional decrease in total precipitation. On the other hand, modern precipitation and temperature records in the region are continuously increasing – at least over the period discussed in the paper. Furthermore, I am discussing the governmental restrictions and the political programme to fight land degradation in northern China. From the vegetation index data, no immediate result can be observed - despite increased precipitation totals and the attempt to mitigate desertification.*

The manuscript is generally well written (although there were few places where I found too long sentences)

*Thanks for that! You were right with the sentences (particularly in the beginning). I changed some of them in the current (potential) revision, which I am preparing.*

and within the scope of the Climate of the Past journal. However, there are several methodological issues, clear hypothesis, linkage between historical data and current land surface conditions and the overall findings of the study. Therefore, substantial revision is necessary prior to improve the manuscript.

1. The author uses four different data sources (historical written sources, paleoenvironmental data, remote sensing products and current gridded

climate data) to find the relationship between the climate data and vegetation indices. It is unclear how the connection between past environmental changes and current land degradation rates, desertification and grassland productivity is made in the study.

While I commend the authors approach and time in analyzing the historical records based on Gerbillon in year 1688, there is no connection made between 1688 and current land degradation rates. The only connection the author made was the linkage between climate data and NDVI for current conditions. I was left wondering whether there is a need to examine the year 1688 (with all the work put forward), which does not add any information to the relationship between climate and NDVI for current conditions.

*Thank you for your comment, but could you specify your concerns a bit more? Do you mean the temporal interval between 1688 and today? As I have pointed out also in the answer to reviewer 2, continuous and extensive surface data is not available, and the spotlight sample locations cannot be interpolated over large areas without creating data narratives. If you mean the conceptual connection between the two chronological periods, I would like to emphasize the potential (as you highlighted) of historical data and written sources to understand landcover-climate feedbacks, which allow to draw conclusions also for current climate change and human impact scenarios. The results I am presenting here are, by nature, based on qualitative analyses for the historical period. If you suggest that historical data analyses or the evaluation of past written sources cannot contribute to current climate change debate, I would like to refer to Historical Geography, Historical Climatology, History itself, and also archaeological research, that significantly contribute to the understanding of past human-landscape and past climate-landcover feedbacks and interactions. The investigation of Pater Gerbillon (subjective because individual) and his perception of the landcover and climate conditions as well as socio-cultural interactions, droughts, crop failure, and livestock perish add an invaluable note to the current understanding of LIA landcover change. And that further allows for the comparison to modern datasets to understand the feedbacks of climate-sensitive regions of the world. Particularly during the LIA, where low temperature and decreased precipitation rates triggered surface degradation, the climatic signal was the major determinant. During current climate change and increasing temperature and precipitation totals, the anthropogenic landcover change is probably the main driving factor of desertification. It is only because of the comparison data that we can further distinguish into an anthropogenic and an environmental component in the signal (and of course they are interlinked!). However, correct me if I am wrong and please add comments to this because this is the most important interface of interdisciplinary research.*

2. The authors conclusion that if there is no environmental and land degradation relationship, current land degradation rates is likely due to intensive livestock grazing needs to be reconsidered. In this study, the authors did not show any relationship between NDVI and livestock grazing, although such data are available at province level from FAO and other sources. While I agree that livestock grazing is likely the cause of current land degradation, there is still a debate on the contribution of land degradation from climate and livestock grazing. It is hard for me to believe that the author used precipitation totals and maximum temperature data to point out no changes in precipitation and an

increase in tmax. The reason is NDVI values does not really work in desert areas or areas that is heavily degraded. Yet, the author tried to establish a relationship between NDVI and climate.

*Please apologize but I am a bit confused with this statement, which probably needs some clarification (?). There is a huge debate in literature, which acknowledges the two potential (and often contradictory) views of a) climate and b) human impact on land degradation processes and I included a review of this literature in the paper. What exactly do you mean with "It is hard for me to believe that the author used precipitation totals and maximum temperature data to point out no changes in precipitation and an increase in tmax"? I described the dataset, cited its origin, and showed the method I used to visualize the data to make it reproducible. I am furthermore not convinced that NDVI is not applicable in degraded or arid environments and there is plenty of research and literature available that use this method to model vegetation response to climate/human impact.*

3. Another issue with the paper is the lack of clear understanding of the contributing factors to desertification or land transformation in Mongolia and Inner Mongolia. The political system of Inner Mongolia and Mongolia have diverged greatly since the collapse of Soviet Union. As a result, land cover change is taking place more rapidly in Inner Mongolia than in Mongolia.

*That is a good point, which I will integrate more intensely in the potential revision of the paper!*

Shifts in policy between inner Mongolia and Mongolia had led to differences in grassland response to climate change and grazing pressure (Chen et al. 2015). I think this should be highlighted in this study and assuming that the Inner Mongolia and Mongolia had similar response to grazing pressure and environmental changes would be questionable given that land response has been increasingly linked to political and policy changes in the region.

*I will include this in the paper with a literature review, thanks for pointing this out to me! My main goal was to show (from a case study) that despite a local increase in total precipitation and the governmental restrictions, no immediate positive vegetation response can be observed. I am sure that both countries experienced different land-use policies during the end of the 20th century and up to now but these differences are not always observable. See e.g. and also for another opinion: (Guo et al. 2021)*

4. It is also unclear how the desertification process or the land degradation rates are estimated in this study. To my understanding, the author is using NDVI values with low NDVI denoting land degradation/desertification. I have serious concern about using NDVI as an indicator of degradation particularly in arid and semi-arid grasslands. I at least want the author to show or cite some previous work that the NDVI can actually detect grassland or ecosystem degradation in arid and semi-arid regions.

*As mentioned earlier, there is plenty of literature and current research – particularly in northern China and Mongolia – that integrates NDVI value analyses and remote sensing techniques in arid and semi-arid landscapes and I have cited some of them. In a potential revision, I will include more recent literature to show the potential of the method.*

5. The relationship between precipitation variability, tmax increase and river runoff is unclear. The author need to justify how increase in tmax and a decline in vegetation cover had no effect on river runoff in the region.

*I am sorry, I do not completely understand this issue. In this paper, reconstructed river runoff derived from palaeoenvironmental proxy based on (Davi et al. 2013; Davi et al. 2006) and I used the data to understand the environmental response to a cooler and drier period during the LIA. Would you mind going into detail here and point out more precisely what you mean? That would be very helpful – thank you!*

Other minor comments

1. Figures: There are 10 figures and a lot of these figures are irrelevant in the main manuscript. For example, Figure 2 is unnecessary since the readers can visually get no information beside the fact that there is no overall change in precipitation between Mongolia and Inner Mongolia. The same applies to other figures.

*I am sorry but I strongly disagree with this statement. Fig. 2 shows precipitation variability in the region and temporal peaks. Furthermore, it shows that there is no severe overall decrease in total precipitation, which is an important information in terms of vegetation response over the past 20 years. Could you please be more precise in your critics and point out which figures you consider irrelevant and unnecessary?*

2. Texts: Some of the text are too long and probably need to be splitted into multiple lines. For example lines 28-30 have too many information, which can easily be divided into multiple lines. I also suggest the author to shorten the text just focusing on what the scope of this paper.

*I will reconsider grammatical issues in a potential revision, depending on the editor's decision.*

3. Lines 19-20: I thought precipitation did not change while there was an expansion of bare lands in modern Mongolia. This lines seems contradictory to the findings

*You were right, there was a word missing, I added 'current' to the following sentence:*
*"However, modern landcover data shows enhanced expansion of bare lands contrasting a current increase in precipitation (Ptotal) and maximum temperature (Tmax)"*

4. Lines 21-22: Can you also add a line on why there was no relationship between Ptotal, tmax and NDVI values? Is it because of the previous year precipitation totals that the current year NDVI is higher? You can easily show this in scatter plot as well. I think climate is still a dominant factor that should define NDVI values given that there are little management activities in the region and given that they practice nomadic pastoralism.

*I am really sorry but line 21-22 are part of the abstract and I cannot add a discussion there about the relationship of the data nor a scatterplot. The discussion part covers these relationships.*

5. Line 42: I am wondering what makes an author really great. Can we just say "emphasized by other studies".

*"A great many" means "a lot" and is not about emphasizing the impact of an author*

6. Line 52: overprint or "footprint"

*That is a good question. I would suggest "overprint" because it emphasizes the totally cultural aspect of what makes a landscape. A footprint would just be another (little) human contribution to global climate change, which I think is misleading…*

7. Lines 96-98: I thought potential land cover maps should have been used here not the current aps.

*Well, potential landcover maps exclude human impact and because this is an important feature of the entire planet, I think the actual maps are suitable for this analysis. I am convinced that potential maps can be misleading due to the fact that the whole region has undergone significant landcover change since the first human-environment interaction and latest since the Neolithic period. We cannot assume 'natural' conditions to model human behaviour in landscapes. That is generally a very interesting discussion and deserves a more methodological approach in another paper.*

8. Lines 138-139: what does this line even mean? Are you implying that the NDVI values in this small section are similar to regional NDVI trend based on MODIS?

*You are right, that was misleading! I deleted this part from the manuscript. These were preprocessing calculations I performed to evaluate the reliability of my data. Thanks for pointing this out to me!*

9. Lines 157-159: I am lost here. How does elevation determine semi-arid conditions? Aridity is a function of precipitation and potential evapotranspiration.

10.

*I am sorry but what do you mean with "you are lost here"? The lines you mention are:*

"Forested zones are mostly abundant in lower elevated areas of the subhumid belt north of Beijing. With increasing elevation, semi-arid conditions prevail, which favour patterns of herbaceous grassland and shrubs. Towards the north-west, extensive sandy lands occur with a mean elevation of about 1000 m a.s.l."

*I do not say that elevation determines semi-arid conditions but that semi-arid conditions prevail with increasing elevation. Correct me if I am wrong because I am not a native English speaker. I am sure that further feedback will enhance the clarity of these issues and foster the overall quality of the general discussion and the paper.*

Chen et al. Policy Shift influence the functional changes of the CNH systems in the Mongolian Plateau. Env Res. Letters 10 085003

**references**

Davi, Nicole K.; Jacoby, Gordon C.; Curtis, A. E.; Baatarbileg, N. (2006): Extension of Drought Records for Central Asia Using Tree Rings: West-Central Mongolia. In: *Journal of Climate* 19 (2), S. 288–299.

Davi, Nicole K.; Pederson, Neil; Leland, Caroline; Nachin, Baatarbileg; Suran, Byambagerel; Jacoby, Gordon C. (2013): Is eastern Mongolia drying? A long-term perspective of a multidecadal trend. In: *Water Resour. Res.* 49 (1), S. 151–158. DOI: 10.1029/2012WR011834.

Guo, Enliang; Wang, Yongfang; Wang, Cailin; Sun, Zhongyi; Bao, Yulong; Mandula, Naren et al. (2021): NDVI Indicates Long-Term Dynamics of Vegetation and Its Driving Forces from Climatic and

Anthropogenic Factors in Mongolian Plateau. In: *Remote Sensing* 13 (4), S. 688. DOI: 10.3390/rs13040688.

---

## Author Comment (AC2)

*Thank you very much for your comments and questions on the manuscript. In the following, I tried to answer most of the issues you raised during the review process. Please find my answers highlighted in italics. I am looking forward to further discussing your interesting suggestions!*

Using historical written sources, palaeoenvironmental data, and Normalized Difference Vegetation Index (NDVI) temporal series to compare landcover change during the Little Ice Age and the reference period 2000-2018, could help us to understand the role of climate in affecting grassland. Surely, you have done a lot of work, but before further consideration, I have several questions at present.

1)" Figure 3 further visualizes the historical route patterns and the daily camps of the travels of Pater Gerbillon from 1688. From these route reconstructions, a cross-validation of hermeneutic sources and modern landcover and climate data was derived." Why modern landcover and climate data was derived from these route reconstructions? In you text, landcover in 2019 was derived from Landsat-OLI-8 satellite imagery. In addition, as we know, from 1688 to now, landcover may be changed many times.

*To evaluate the historical landcover in terms of climate extremes during the LIA, I need to contextualize the reconstructed landcover classification from the year 1688. Of course, you are totally right in emphasizing the potential landcover transformation during the period 1688-2000. But, you know, palaeoenvironmental data or historical landcover data covering such a broad region is hardly available and cannot be interpolated from scattered proxies without risking massive loss of information and creating strongly hypothetical data in peripheral areas that were lacking samples. Because I am focussing mostly on the comparison of two climate extremes (LIA and current climate change) and I want to cross-validate vegetation response during cold/dry and hot/more humid conditions, I would like to contrast the modern surface cover to the historical data.*

2)"Climatic conditions were reported to be very dry and extremely cold during October 1688, which aligns with the climatic tendency towards a drier and colder period around 1700 AD and the climate depression during the Maunder Minimum of the LIA. From the palaeoenvironmental reconstructions, 1688 can be considered an extremely anomalous year compared to the long-term average and marks the transition into a generally colder and drier phase that lasts until about 1715 AD." From this, climate of 1688 is clear, and then is it necessary to reconstruct?

*I am sorry, maybe I do not understand this point properly. If I understand correctly, you suggest that a reconstruction of climate and surface conditions during the LIA is not necessary because we know from literature that this happened? I would consider this a dangerous perspective not only in terms of scientific paradigms and research approaches in general but also because local and regional response to climate change (also in historical times) is yet not fully understood and as mentioned above, micro-regional or site-specific reconstructions from palaeoenvironmental proxy can hardly be interpolated to supraregional premodern conditions.*

*If I misunderstood your point, I am looking forward to further discussion about this!*

3) In aird and semi-arid area, plant growth may be more close to heavy precipitation events but not to total precipitation as presented in Figure 9.

*That is a very good point, which I will highlight more accurately in the paper, thank you for this. Also, I think that heavy precipitation can cause massive surface erosion and soil loss and thus amplifies the previous hot drought period and contributes to degradation processes. This, however, is also closely linked to terrain roughness and potential erosion.*

4ï¼‰"According to written sources, the year was characterized by extremely low temperatures during late grazing season of September and the onset of October and extremely dry conditions and severely high temperatures during summer rainy season, which caused massive perish of livestock in the region." From what you can say 'perish'?

*This is information can be extracted from the diary by Gerbillon, who documented a very strong perish of livestock during his travel.*

5) "Around the reconstructed route, a 20 km buffer was created to visualize the historical environmental conditions within a suitable range". Surface conditions are various under various toporograph. For example, with 20 km buffer, there are various landscapes, such as cropland, forest, grassland, shrubland, and waters. Can surface conditons from camps extend to 20 km?

*That is an interesting point, which is discussed in manifold theoretical discourses from both the sciences and the humanities. The actual 'catchment' of a site, or in this case, the scale of perception of the individual, is to a large extent biased by the performance of the individual itself. And that accounts also for potential viewshed, current physical condition, or weather conditions and time of the day. A 20 km range produces a large buffer, however, a 10 km range would produce equal results but in a smaller buffer. The reconstructions, which I would like to suggest here, are at best a realistic model of a potential past and never a reconstruction per se. A model of the landscape, in the very sense of being a simplification of characteristic parameters regarding some potential landscape variables, is not affected by the diameter of the range. It produces a stripe of landscape, which for visualization purposes, can serve as comparison data but never as a real reconstruction of past conditions.*

6) If we don't know the number of population and livestock around 1688, the human distrubances on grassland degradation can not be well understood. Climate change is only one reason that can explain grassland succession.

*That is definitely true and one weak point in any model that aims at tracing past landcover change as result of past human behaviour. In historical and archaeological*

*research, demographic numbers or the number of cattle in specific areas cannot be reconstructed with absolute certainty. However, just like in the concept of modelling past human-landscape interactions, we have to deal with the data that is available and probably this is not always accurate. And of course, climate change is one potential parameter during degradation processes but the amplification through human pressure on available resources (such as grasslands) can be considered a major trigger of extensive desertification – particularly in sensitive steppe vegetation regions.*

7) "Results show that decreasing precipitation and temperature records led to increased land degradation during the late 17 th century". Does this mean land degradation occure the late 17th century?  "no major grassland recovery over the past 20 years", the reference undegradation year is?

*Please apologize but I am a bit confused by this point. Do you mean that by the end of the 17th century, climate deterioration occurred? If this is the case, I would say yes. We can see from the palaeoenvironmental models that there is a significant decline in temperature and a decreasing trend in precipitation and river runoff. That aligns with climate trends during the LIA and a general trend towards dry and cold conditions.*

*I am very sorry, but what exactly do you mean with "the reference undegradation year is"? You seem to refer to a section of the abstract, which summarizes the results from the modern comparison NDVI dataset. The references for that can be found in the text.*

---

## Author Comment (AC4)

Reply to comments: **RC3**: , Anonymous Referee #2, 14 Apr 2021

Many thanks for your comments! Please find my reply in italics following your original question

1 From these route reconstructions, a cross-validation of hermeneutic sources and modern landcover and climate data was derived.

Here, do you mean that by cross-validation, palaeo-climate and current climate data, and historical surface cover and modern surface cover data, were got, respectively?

*Yes, that is basically what I meant. The reconstructed route is based on the diary and serves as basis from which the (almost) daily palaeoenvironmental information was extracted. That includes a temporal axis and not only provides spatial data. That is why the temporal shift in the data needs to be considered as well.*

As I know, from route reconstructions, by cross-validation you can get data of palaeo-climate and historical surface cover. Otherwise, I missed something.

*Yes, that is what I did…*

2 Results show that decreasing precipitation and temperature records led to increased land degradation during the late 17 th century.

Does this sentence mean that land degradation has occurred the late 17th century?

*Yes, that is totally what it says.*

---

## Author Comment (AC5)

Changs of plant morphology to adapt environmental change such as decreasing precipitation and temperature can be seen as 'degradation'? As I think, environmental issues caused by human can be regarded 'degradation'. Otherwise, they are natural laws.

*You are totally right about the terminology! But now, when I am thinking of the temporal frame of the period under consideration and the already high anthropogenic overprint in the region (and the world basically), it would also be a question whether environmental signals can be considered separate from human-induced changes. I mean, considering human impact since (at least) the Neolithic and the palimpsest-like construction of 'cultural landscapes', which is built on massive land-use of prior social groups, to which extent can we state that there is purely environmental change? This is, of course, also a methodological issue in both, the sciences and the humanities…*

Land degradation occured during the later 17 century. This statement is lack of evidence, and perish of livestock may be too weak.

*I can only tell from the historical data… the diary states that there is considerable loss of livestock and that cold and dry conditions prevailed. It also says that these climatic conditions and the consequent surface transformation is totally unusual, which, as you say, is of course completely subjective and represents an individual perception of the region. But this is the data – and that is what we have to deal with.*

If land degradation has occured since that time, servious degradation has been continuous till now as population and economy growth even if cliamte becomes better in the study area. Of course, due to lack historical data, somethings are easy disputable.

*I am convinced that exactly this happened: since the 17th century and throughout the modern era, massive degradation as occurred, mostly driven by human overstraining of the regionally very sensitive vegetation. Also, resource exploitation needs to be considered, which has caused increasing loss of groundwater. To which extent this has changed recently due to China's anti-desertification programme, needs to be more precisely considered! I am right now preparing another dataset, from which these hypotheses can be more clearly discussed. As soon as I got the results, I will upload them to the discussion!!*

---

## Author Comment (AC7)

**Author correction to the article**

Michael Kempf (2021)

**Monitoring landcover change and desertification processes in northern China and Mongolia using historical written sources and vegetation indices**

Current climate change and precipitation and temperature anomalies in Northern China and Mongolia were plotted and compared to vegetation indices (MODIS/Terra Vegetation Indices Monthly L3 Global 0.05Deg CMG V006; last accessed 07th of April 2021; https://lpdaac.usgs.gov/products/mod13c2v006/) (CRU, Harris and Jones, 2019, Harris).

The datasets were reprojected to WGS84 (EPSG4326) using the warp (reproject) tool in QGIS. A point dataset with 1000 random points was created within a 20 km buffer around the reconstructed route from 1688. Using the point sampling tool plugin in QGIS, each monthly environmental variable was added to each point and the mean values were calculated. Using the R package *ggpubr* developed by Alboukadel Kassambara and this R code

```
**read data**
data <- read.csv("mydata.csv", header = TRUE)
**Perform Pearson's correlation**
ggscatter(data, x = "mydata1", y = "mydata2", add = "reg.line", conf.int =
TRUE,
        col="black",  cor.coef = TRUE,
    cor.method = "pearson",
     xlab = "mydata1", ylab = "mydata2")
```

a Pearson correlation test was performed. Result show that there is a strong correlation between total vegetation growth behavior and precipitation and Tmax respectively. The correlation test, however, has been performed including the mean values of the random point samples, which covered the entire route and not, like in the preprint version of this article, just a small section in Northern China's Inner Mongolia during the early growing season (March - May).

If the paper is further considered for publication, the new evidence and the results will be included and discussed in the revised version of the paper.

Please find the diagrams of precipitation, Tmax, and NDVI variability as well as the correlation plots attached to this correction manuscript (Fig. 1-5).

[Figure]

Fig. 1: Total NDVI mean values for the period Feb. 2000 – Dec. 2018 based on 1000 random points distributed in a 20 km buffer section along the reconstructed route from 1688. The values range from 227 (Feb. 2000) to 0 (Dec. 2018) and show a significant increase in physical plant condition. Uncalibrated NDVI data based on MODIS/Terra Vegetation Indices Monthly L3 Global 0.05Deg CMG V006; last accessed 07th of April 2021; https://lpdaac.usgs.gov/products/mod13c2v006/.

[Figure]

Fig. 2: Total precipitation mean values for the period Feb. 2000 – Dec. 2018 based on 1000 random points distributed in a 20 km buffer section along the reconstructed route from 1688. The values range from 227 (Feb. 2000) to 0 (Dec. 2018) and are based on (CRU, Harris and Jones, 2019). Total precipitation increases from 2000 to 2018.

[Figure]

Fig. 3: Total Tmax mean values for the period Feb. 2000 – Dec. 2018 based on 1000 random points distributed in a 20 km buffer section along the reconstructed route from 1688. The values range from 227 (Feb. 2000) to 0 (Dec. 2018) and are based on (CRU, Harris and Jones, 2019). There is an increase in Tmax from 2000 to 2018.

[Figure]

Fig. 4: Pearson's correlation test to estimate the relationship between precipitation and vegetation response. Vegetation response is significantly correlated to an increase in precipitation.

[Figure]

Fig. 4: Pearson's correlation test to estimate the relationship between Tmax and vegetation response. Vegetation response is significantly correlated to an increase in Tmax.

**Reference list**

University Of East Anglia Climatic Research Unit (CRU), Harris, I.C. and Jones, P.D. (2019) *CRU TS4.03: Climatic Research Unit (CRU) Time-Series (TS) version 4.03 of high-resolution gridded data of month-by-month variation in climate (Jan. 1901- Dec. 2018)*.